# Crystal structures of human MGST2 reveal synchronized conformational changes regulating catalysis

Madhuranayaki Thulasingam [1✉], Laura Orellana[2,3], Emmanuel Nji [2,4], Shabbir Ahmad [1,5], Agnes Rinaldo-Matthis[1] & Jesper Z. Haeggström[1✉]

Microsomal glutathione S-transferase 2 (MGST2) produces leukotriene $C_4$, key for intracrine signaling of endoplasmic reticulum (ER) stress, oxidative DNA damage and cell death. MGST2 trimer restricts catalysis to only one out of three active sites at a time, but the molecular basis is unknown. Here, we present crystal structures of human MGST2 combined with biochemical and computational evidence for a concerted mechanism, involving local unfolding coupled to global conformational changes that regulate catalysis. Furthermore, synchronized changes in the biconical central pore modulate the hydrophobicity and control solvent influx to optimize reaction conditions at the active site. These unique mechanistic insights pertain to other, structurally related, drug targets.

[1] Department of Medical Biochemistry and Biophysics, Division of Chemistry II, Karolinska Institutet, Solnavägen 9, 171 65 Stockholm, Sweden. [2] Department of Biochemistry and Biophysics, Stockholm University, Svante Arrhenius väg 16, 106 91 Stockholm, Sweden. [3] Present address: Department of Oncology-Pathology, Karolinska Institutet, Stockholm, Sweden. [4] Present address: BioStruct-Africa, Stockholm, Sweden. [5] Present address: Department of Cell and Molecular Biology, Uppsala University, Uppsala, Sweden. ✉email: madhuranayaki.thulasingam@ki.se; Jesper.Haeggstrom@ki.se

Microsomal glutathione S-transferase 2 (MGST2) is a member of a family of integral membrane proteins denoted MAPEG (Membrane Associated Proteins in Eicosanoid and Glutathione metabolism), which encompasses several important targets for development of anti-inflammatory and anti-cancer drugs interfering with prostaglandin and leukotriene biosynthesis[1,2]. In non-hematopoietic cells, MGST2 catalyzes leukotriene $C_4$ ($LTC_4$) synthesis via conjugation of glutathione (GSH) and $LTA_4$, a transient epoxide intermediate derived from 5-lipoxygenase metabolism of arachidonic acid[3,4]. In such cells, $LTC_4$ functions as an intracrine mediator of cell death signaled by endoplasmic reticulum stress and oxidative DNA damage induced by common chemotherapeutic agents[5]. MGST2 can also produce cysteinyl immunoresolvents, a recently discovered family of specialized proresolving mediators, and exhibits a GSH-dependent lipid peroxidase activity[6,7]. Previous studies of MGST2 have revealed that only one out of three active sites is used at a time, an intriguing catalytic behavior also observed for MGST1 and microsomal prostaglandin E synthase-1 (mPGES-1) both of which are strongly implicated in human pathophysiology[7–9].

Here we determined three crystal structures of human MGST2: holoenzyme with GSH, apoenzyme, and a complex with the substrate analog/inhibitor glutathione sulfonic acid ($GSO_3^-$), which together with biochemical and computational evidence reveal the molecular basis for one-third the sites reactivity of MGST2 catalytic mechanism.

## Results and discussion

### MGST2 exhibits unique flexibility and asymmetric co-substrate binding.

The overall structure is a homo-trimer, with each monomer containing 4 trans-membrane α-helices (αH1–αH4), forming three potential active sites at their interfaces (Fig. 1a, b). Although showing significant disorder in the crystal structures at the terminal ends, MGST2 shows the beginning of a C-terminal extra-membrane αH5 (Fig. 1a and Supplementary Fig. 1a) and a structurally conserved "loop L" connecting αH1 to αH2[10] (Fig. 1a and Supplementary Fig. 1b). The N and C termini were experimentally shown to be located within the lumen (Supplementary Fig. 2), which is opposite to what has been reported for $LTC_4$ synthase[11]. The protomers are held together by extensive hydrogen, hydrophobic, and polar interactions. Residues from αH1, αH2, αH4, and loop L but not from αH3 are involved in the inter-subunit interactions. Specifically, Glu58 on αH2 provides a major symmetric polar interaction with Gln53 on αH2 of opposite monomer (Supplementary Fig. 3a–c). We find that MGST2 exhibits an unusual flexibility and asymmetry. The structure of apo-MGST2 (with two homo-trimers in the asymmetric unit) shows structural heterogeneity (Supplementary Fig. 3d), while the structure of holo-MGST2, which contains GSH at only two of the three possible active sites appears more rigid (Fig. 1b). In contrast to the holo complex, MGST2 co-crystallized with the inhibitor $GSO_3^-$ reveals full occupation of all three active sites (Fig. 1c, d and Supplementary Fig. 4). Comparison of the apo and holo structures uncovers distinct local structural changes associated with GSH binding that are accompanied by a global asymmetry previously unobserved within the MAPEG family[10,12–14].

In the holo structure, only one of the GSH molecules is observed in full occupancy, with the other refined to 72%. The binding conformation of GSH as well as those of associated interactions with active site residues differs markedly between these two sites (Fig. 2a–c), suggesting that only the GSH at full occupancy is bound productively. Furthermore, the GSH thiol interacts with Arg104, which upon mutation to Ala or Lys renders MGST2 inactive (Fig. 3a) and unable to generate a thiolate signal at 239 nm (Fig. 3b, Supplementary Fig. 5, and

Supplementary Table 1). Thus, Arg104 is critical for thiolate anion formation and stabilization and is key to the enzyme's $LTC_4$ synthase activity, in agreement with previous data for MAPEG members[15,16]. Of note, the binding conformation of $GSO_3^-$ is different to that observed for the fully occupied GSH in the holo complex (Fig. 2d and Supplementary Fig. 4), suggesting that interaction with the unmodified thiol is essential for productive binding of GSH.

The GSH at full occupancy observed in the holo complex adopts a "crescent"-shaped binding mode (Fig. 2a), similar to what has been reported for $LTC_4$ synthase[10,11] and mPGES-1[12], while contrasting to the extended conformation described for MGST1[17,18]. In comparison, the GSH at partial occupancy displays several distinct features. This GSH has an opposite head to tail orientation with its γ-glutamyl amino group hydrogen bonded to Tyr97 instead of Glu58 as for the GSH at full occupancy (Fig. 2b, c). Mutation of Glu58 to Ala resulted in 93% reduction in $LTC_4$ synthase activity, whereas mutation of Tyr97 to Phe maintained 26% of the activity, which substantiates the role of Glu58 in effective GSH binding (Fig. 3a). Furthermore, the salt bridge between Arg51 and the glycyl carboxylate of GSH is lost at the partially occupied site, which contributes to its inefficient binding. This shift will likely affect catalysis since mutation of Arg51 into Ala or Lys leads to complete or near complete loss of enzyme activity, corroborating its critical role in productive GSH binding (Fig. 3a).

### Local and global conformational changes of MGST2 are associated with GSH binding.

Intra- and inter-monomeric conformational changes accompany the local asymmetry specific to effective GSH binding and subsequent catalysis at only one site at a time. Thus, GSH binding is associated with remodeling of nearby polar interactions. Notably, Asn55 on αH2 forms an intra-monomeric interaction with Arg90 on αH3 in the apo structure while it moves toward GSH in the holo structure (Fig. 4a–c). Besides the interaction with GSH, Asn55 also coordinates with Arg51 on αH2 of the same monomer at the fully occupied site and this coordination is not observed in the other two monomers (Fig. 4a). Furthermore, relative to the apo structure, Glu58 on αH2 switches its inter-monomeric interaction from Gln53 to Gln54 on the αH2 of opposing monomer to form a salt bridge with the γ-glutamyl terminal amine of the productively bound GSH. This Glu58–Gln54 interaction is also present at the empty site, whereas the partially occupied site still maintains the Glu58–Gln53 interaction (Fig. 4d).

The local changes associated with GSH binding coincide with shifts of the secondary and tertiary structure near the unoccupied active site of the holo complex. Specifically, the four-residue stretch (Ala101–Arg104) at the beginning of αH4, which forms part of the active site and contains the catalytically important Arg104, is uniformly observed in a $3_{10}$ helical motif within the apo structure and the $GSO_3^-$ complex (Supplementary Fig. 3 and Fig. 1c, d). In the holo–GSH complex, however, this region adopts a loop conformation with relatively high B-factor within the unoccupied active site (Fig. 5a, b). This transition from $3_{10}$ helix to loop makes the essential Arg104 more flexible, which is reflected in poor electron density compared with GSH-bound sites (Supplementary Fig. 6). Moreover, structural rearrangements at this site also induce a tilt of αH4 of ~5° that results in a significant displacement of the corresponding N-terminus of αH4 as compared to the apo structure (Supplementary Fig. 7).

To probe the dynamics of the GSH-binding site, we simulated the apo and holo conformations both in the absence and presence of GSH. In apo simulations on a membrane, the symmetry of inter-subunit interactions early breaks and the $3_{10}$ helix at the

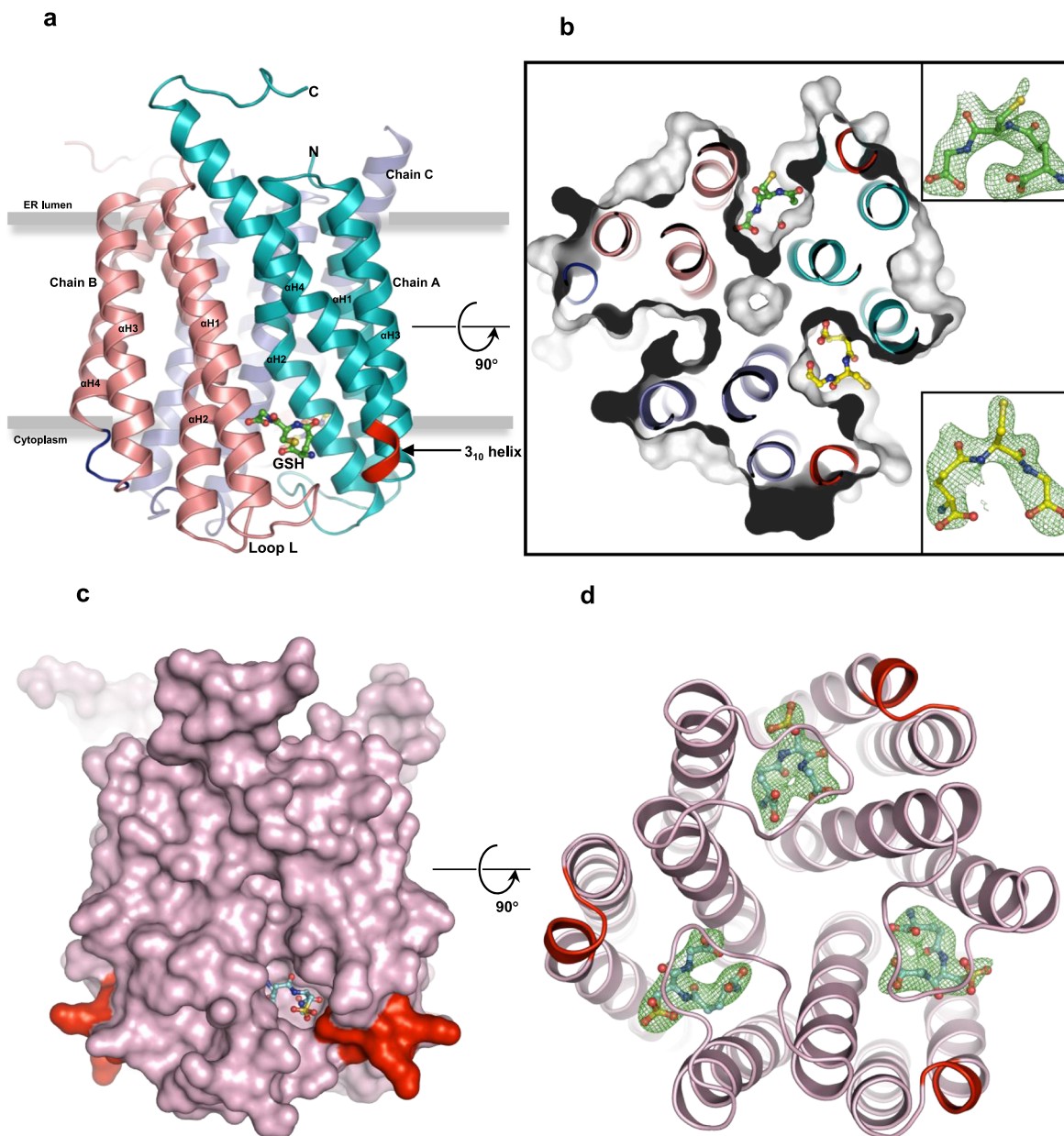

**Fig. 1 Architecture of MGST2. a** Cartoon rendering of trimeric holo-MGST2 structure viewed from the membrane plane. Monomers of the trimer are colored in teal (chain A), salmon (chain B), and slate (chain C). **b** Cross-section from the cytoplasm displays asymmetric binding of GSH at the dimer interface in surface rendering. GSH at full occupancy (yellow ball and stick) and partial occupancy (green ball and stick) shown in Polder OMIT map contoured to 3σ in the insets. **c** Surface rendering of MGST2-GSO$_3^-$ complex structure, bound with 3 molecules of GSO$_3^-$ in each of the active sites and **d** view from the cytoplasm with GSO$_3^-$ molecules in Polder OMIT map contoured to 3.0σ. The 3$_{10}$ helical motif is highlighted in red.

active site unfolds in one unit at a time (Fig. 5c and Supplementary Movie 1). Moreover, such unfolding is also reversible with some runs undergoing transient refolding. This local unfolding is not subunit-specific, occurring in a different monomer in each replica trajectory, and is accompanied by changes in inter-subunit contacts. While Glu58 interacts by forming H-bonds with both Gln53 and Gln54 in all the interfaces, only in the unfolded subunit it forms a transient salt bridge with Arg30 of the neighboring one, bringing closer GSH-binding site residues (Supplementary Fig. 8a). On the contrary, in spite of considerable global flexibility, the 3$_{10}$ helix appears remarkably stable in the holo conformer even after GSH removal (Supplementary Fig. 8b). In each of the apo trajectories, one monomer has a perfectly folded site, another a partially folded, and the last

is fully unfolded (Fig. 5c). These asymmetries appear also in the holo state, in which the GSH at partial occupancy heavily fluctuates and finally leaves the active site via loop L opening while the fully occupied stays tightly bound (Supplementary Movie 2). Notably, such unfolding significantly increases solvent accessibility to the binding site, thus facilitating the entry of the hydrophilic substrate GSH and its interaction with Arg104, entirely hydrated in a water pocket (Supplementary Fig. 9). Globally, the apo simulations spontaneously approach the holo state and vice versa through large-scale changes like αH4 tilting (Supplementary Fig. 10), which are accompanied locally by the highly dynamic equilibrium of the 3$_{10}$ helix unfolding and refolding at one unit at a time and other changes roughly preconfiguring a binding site. These evidences suggest an

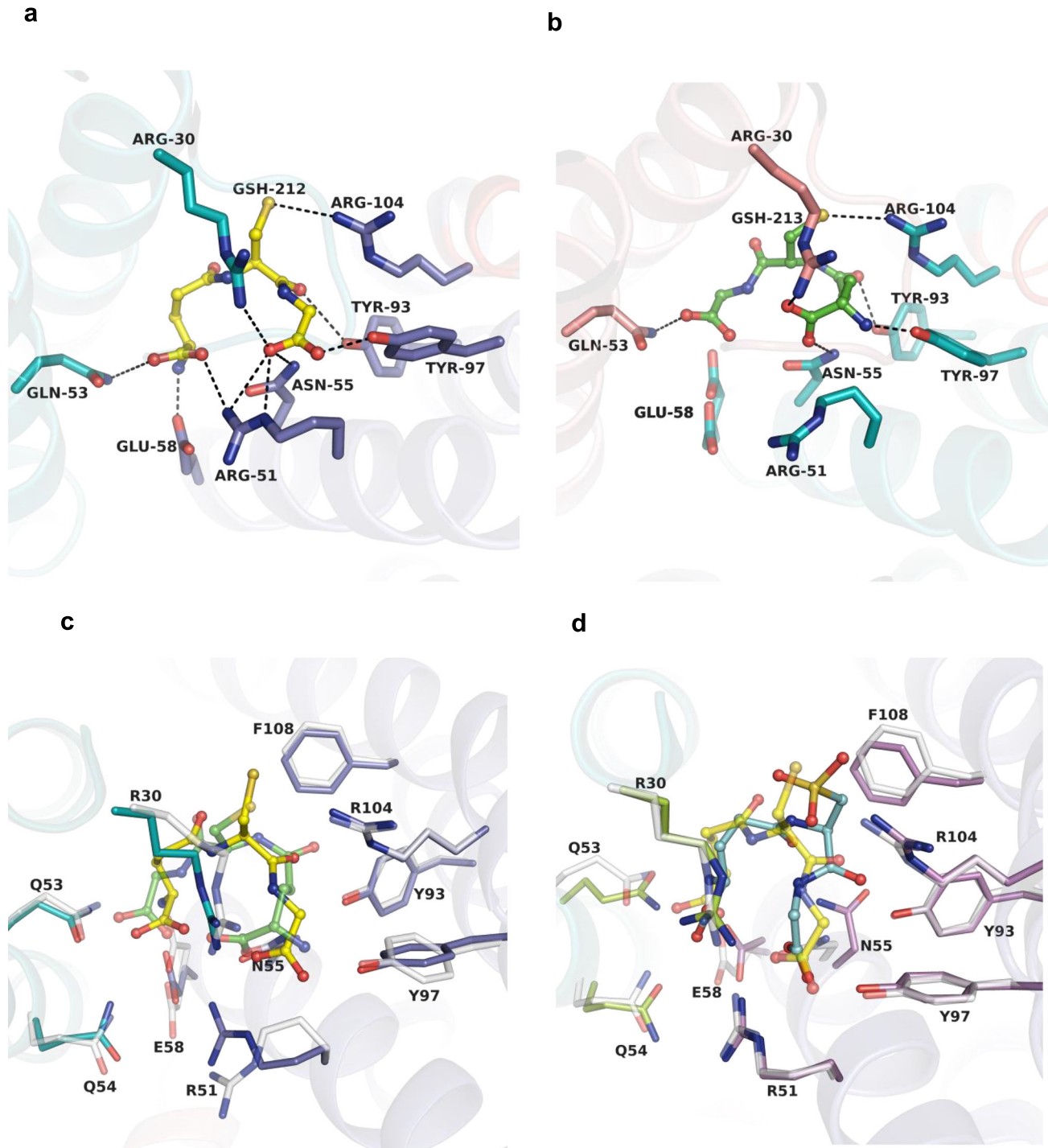

**Fig. 2 Binding mode of GSH and GSO$_3^-$.** The GSH bound at the dimer interface establishes chemical interactions (dashed lines) with residues from αH2, αH3, and αH4 of one monomer and αH1 and αH2 of opposite monomer. **a** Interaction of GSH (yellow ball and stick) at full occupancy site and **b** GSH at partial occupancy (green ball and stick) with key active site residues of corresponding monomers. **c** Superposition of GSH conformations in holo-MGST2 at the active site with GSH bound at full occupancy (opaque) and partial occupancy (translucent). **d** Superposition of active sites of MGST2-GSO$_3^-$ complex (opaque) and holo-MGST2 with GSH at full occupancy (translucent).

important role for conformational selection in the initial steps of MGST2 catalytic mechanism, which warrants further investigation.

**Conformational changes at the luminal and cytoplasmic sides of MGST2 can control solvent access and catalysis.** Crystal

structures of apo- and holo-MGST2 also revealed substantial changes along a central channel of the enzyme, formed by the trilateral orientations of αH2 (Fig. 6a). A conserved Pro61 induces a kink approximately in the middle of this α-helix, creating a constricted "bottleneck" in the membrane-spanning cavity, thus imparting its bi-conical "hour-glass" shape (Fig. 6b), also noted in other MAPEG members[12]. However, we observed

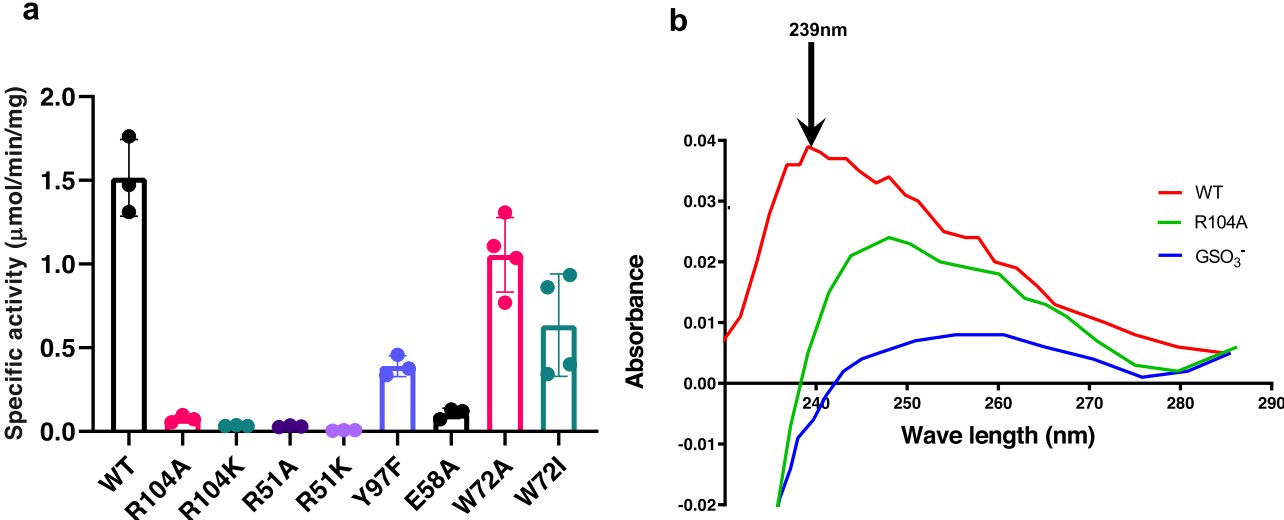

**Fig. 3 Mutational analysis of MGST2. a** Wild type and mutant enzymes were assayed for LTC$_4$ synthase activity. Error bars represent the mean values ± standard deviations of three to four independent activity measurements. **b** Thiolate anion formation in MGST2: UV difference spectra recorded from 230 to 285 nm demonstrate the thiolate anion peak at 239 nm, which is absent in Arg104Ala mutant and in the presence of GSO$_3^-$ inhibitor. Source data are provided as a Source data file.

significant structural differences in pore-lining residues between the apo and holo states: for instance, reorientation in the side chain of Val57 upon GSH binding broadens the cytoplasmic side of the cavity possibly increasing solvent access (Fig. 6c). At the luminal side, GSH binding induces a stabilizing transition in the opposing rotameric positions of Trp72, forming a "tryptophan-gate" (Trp-gate) that could regulate solvent accessibility (Fig. 6b). Simulations coupled these structural changes to striking differences in the flexibility of MGST2 luminal and cytosolic sides (Supplementary Fig. 11), which heavily determine pore access and volume. As the central channel is reshaped depending on GSH absence or presence (Supplementary Fig. 12), MGST2 arrangement on the membrane changes in parallel, alternatively exposing or burying the cytosolic side of the enzyme (Supplementary Fig. 13). This notion is further supported by limited trypsin digestion, where the enzyme is less susceptible to proteolytic cleavage of dynamic and surface-exposed loop L and 3$_{10}$ helix in the presence of GSH (Fig. 6d).

In the apo state, 3$_{10}$ helix unfolding coupled with αH4 tilting is accompanied by further structural changes: loop L opens, Pro61 kink unfolds, the kink angle increases, and Trp-gate opens (Figs. 6e and 7a). These conformational changes widen the central pore, allow a significant water influx throughout the inner cavity (Fig. 7b and Supplementary Fig. 14), and expose the active site to solvent. In this state, the highly dynamic loop L, which protrudes to the solvent, can trap GSH and help approximate it to the binding site (Fig. 7e and Supplementary Movie 3). In contrast, in the holo state, the productively bound GSH interacts tightly within the active site, which globally rigidifies MGST2. This produces a hydrophobic, ordered conformation of the entire central channel and restricts solvent entry from both the luminal and cytosolic side (Fig. 7c, d and Supplementary Fig. 14), which facilitate the hydrophobic second substrate entry and subsequent catalysis as schematically depicted (Fig. 7e). Moreover, mutational analysis revealed that Trp72Ala and Trp72Ile isoforms, where Trp-gating is impaired, exhibit an attenuated LTC$_4$ synthase activity of 25–50% relative to that of wild-type enzyme (Fig. 3a). This is remarkable considering that these residues are located far away from the catalytic site. The observed differential dynamics at the cytosolic and luminal sides of MGST2 resembles

mechanisms observed in transporters to alternate ligand access to either side of the membrane[19]. One may speculate that they reflect a potential mechanism of regulated bipartite solvent access to the active site of MGST2 that may serve to facilitate catalysis and/or entrance of lipid substrates as well as exit of amphipathic or peroxidase products.

Generation of LTC$_4$ by MGST2 was recently reported as an important intracrine mechanism of signaling necroptosis in non-hematopoietic cells. It appears likely that such a process requires tight regulation of the MGST2 catalytic machinery. The first steps are chemically challenging, involving conjugation of a hydrophilic (GSH) and a hydrophobic unstable substrate (LTA$_4$) to generate the amphipathic product (LTC$_4$). Our results provide evidence that regulation is achieved by a structurally dynamic mechanism, which allows switching the hydrophobicity of MGST2 central cavity for LTC$_4$ production, at the same time that access to the binding site is restricted to only one out of three active sites at a time, as schematically depicted in Fig. 7e. Our work will specifically help design small molecules inhibiting MGST2, which may reduce toxic side effects of common chemotherapeutic agents. Moreover, these structural and mechanistic insights also pertain to other MAPEG members some of which are important targets for development of anti-inflammatory drugs.

## Methods

**Protein expression and purification**. Human *Mgst2* cDNA (IMAGE cDNA clone 5277851, Medical Research Council gene service, Cambridge, UK) was cloned in pPICZA vector (Invitrogen) with N-terminal His-tag and transformed into *Pichia pastoris* KM71H cells using the Pichia EasyComp Transformation Kit (Invitrogen). The recombinant cells were grown in 4 l of buffered minimal yeast medium containing glycerol (Invitrogen formula) at 27 °C in baffled flasks. The cells were harvested when OD 600 reached 8–10 and resuspended in 2 l of buffered minimal yeast medium, supplemented every 24 h with 0.6% (v/v) methanol and the pH of the medium was adjusted to 6–6.5 using 8% (v/v) NH$_3$. After 48 h, cells were harvested and resuspended in 50 mM Tris-HCl pH 7.8, 100 mM KCl, and 10% (v/v) glycerol. Homogenization was carried out with glass beads in Bead Beater (BioSpec Products, Inc.) for 9× 1 min with 7-min cooling break between each cycle. The homogenate was filtered through nylon net filter and centrifuged at 1500 × *g* for 10 min to remove unbroken cells and debris. The supernatant containing the membrane fraction was solubilized at 4 °C for 1 h along with 1% (w/v) Triton X-100, 0.5% (w/v) sodium deoxycholate and 5 mM 2-mercaptoethanol under constant stirring. Solubilized content was centrifuged at 10,000 × *g* for 10 min. In all, 10 mM imidazole was added to the supernatant and loaded on to Ni-Sepharose Fast Flow (GE Healthcare) column. The column was washed with five column

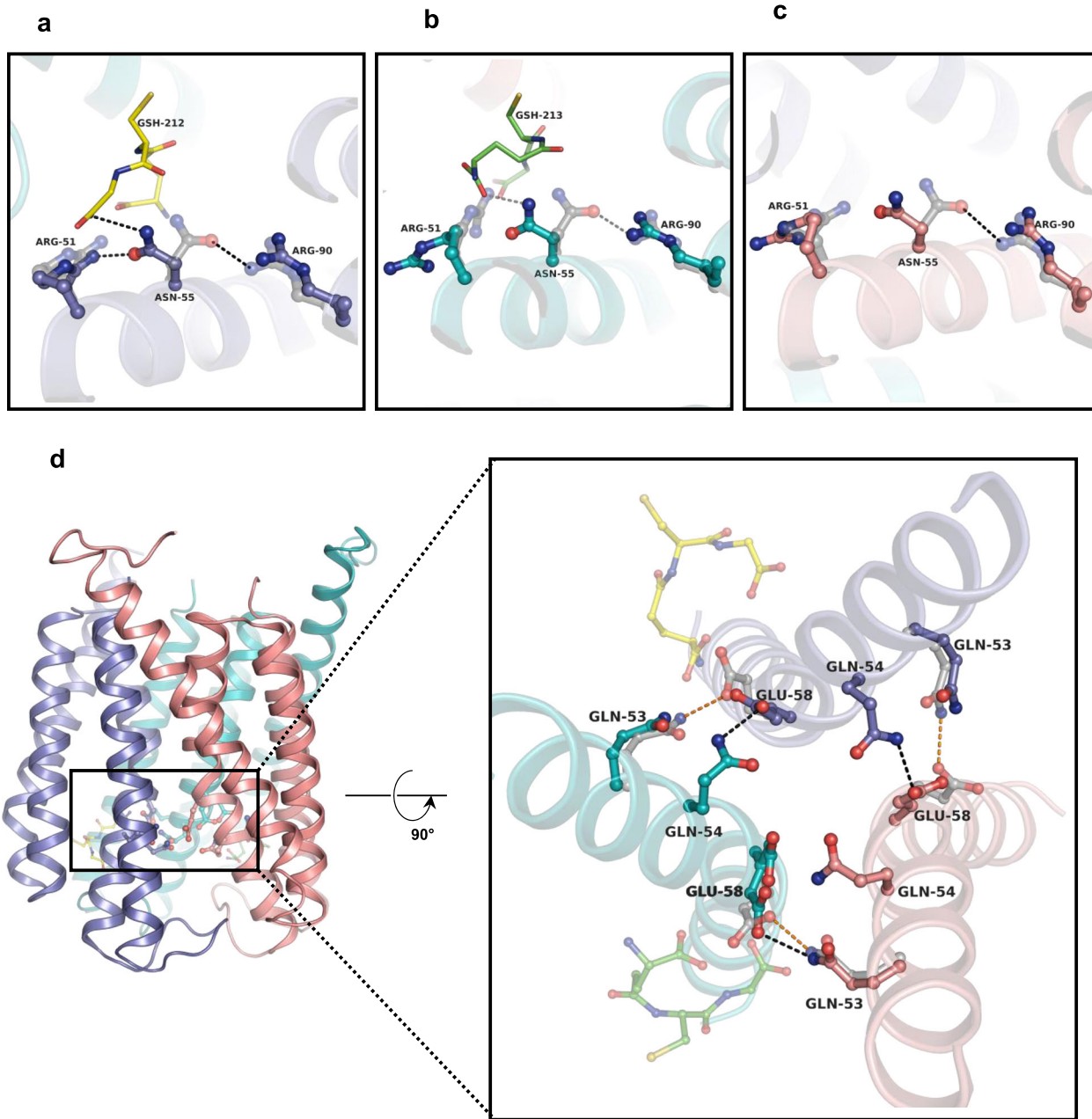

**Fig. 4 Alteration of intra- and inter-monomeric interactions accompanied by GSH binding.** Superimposed apo (gray translucent ball and stick) and holo-MGST2 (opaque ball and stick) structures displaying the reshaping of Asn55-Arg90 intra-monomeric interaction: **a** active site with GSH at full occupancy, **b** active site with GSH at partial occupancy, and **c** active site without GSH. GSH molecules are shown in stick representation. **d** Asymmetric inter-monomeric interaction between αH2 (cartoon): Glu58 switches its interaction from Glu53 to Gln54 at the GSH unoccupied and fully occupied sites, whereas Glu58 remains interacting with Gln53 at the partially occupied site as observed in the apo structure. Apo residues and interactions are shown in gray translucent ball and stick and orange dashed lines, respectively.

volumes (CV) of buffer containing 25 mM Tris-HCl pH 7.8, 500 mM NaCl, 10% (v/v) glycerol, 5 mM 2-mercaptoethanol, 0.05% (w/v) *n*-dodecyl β-D-maltoside (DDM), and 20 mM imidazole followed by three CV of same buffer having 40 mM imidazole instead of 20 mM concentration. Protein was eluted with elution buffer containing 25 mM Tris-HCl pH 7.8, 500 mM NaCl, 10% (v/v) glycerol, 5 mM 2-mercaptoethanol, 0.05% (w/v) DDM, and 300 mM imidazole. The protein used for activity measurement was eluted without GSH (Sigma) in elution buffer and further buffer exchanged to 25 mM Tris-HCl pH 7.8, 100 mM NaCl, 10% (v/v) glycerol, 5 mM 2-mercaptoethanol, and 0.05% (w/v) DDM using PD-10 columns (GE Healthcare).

MGST2 protein used for structural studies was eluted with 0.1 mM GSH in the Ni-Sepharose elution buffer and the fractions were pooled and passed through *S*-hexylglutathione agarose column (Abcam and GE Healthcare). Column was washed with three CV of buffer containing 25 mM Tris-HCl pH 8.0, 500 mM NaCl, 10% (v/v) glycerol, 5 mM 2-mercaptoethanol, 0.05% (w/v) DDM, and 0.1 mM GSH. Protein was eluted with five CV of 25 mM Tris-HCl pH 8.0, 10% (v/v) glycerol, 5 mM 2-mercaptoethanol, 0.05% (w/v) DDM, 0.1 mM GSH, and 30 mM probenecid. The eluted protein was concentrated using Amicon Ultra 30-kDa cutoff centrifugal filter device (Millipore) and purified further by size exclusion chromatography (SEC) using Superdex 200 16/600 (GE Healthcare) column. Final buffer used at SEC contains 25 mM Tris-HCl pH 8.0, 100 mM NaCl, 10% (v/v) glycerol, 0.1 mM tris(2-carboxyethyl)phosphine (TCEP), 0.03% (w/v) DDM, and 0.1 mM GSH. SEC fractions containing pure MGST2 was concentrated to 30 mg ml$^{-1}$ by ultrafiltration and stored frozen at −80 °C freezer. Protein concentration was measured by ultraviolet (UV) spectrophotometry, and purity was analyzed by sodium dodecyl sulfate–polyacrylamide gel electrophoresis (SDS-PAGE).

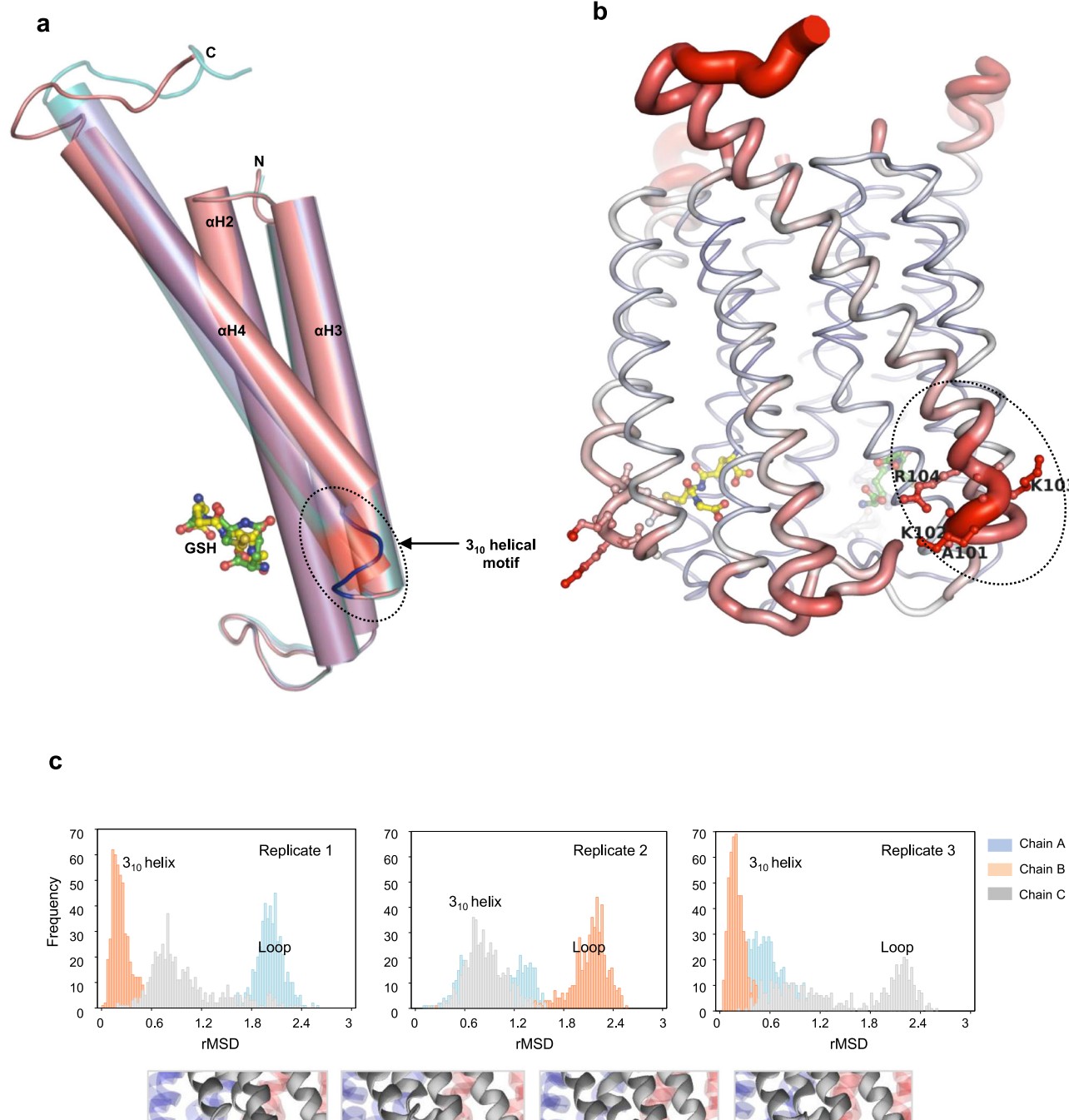

**Fig. 5 Structural asymmetry of MGST2. a** Superimposition of chain A (translucent teal) and chain C (translucent slate) on chain B (opaque salmon) of holo-MGST2 in cylindrical rendering. Note the loop conformation (blue) of the $3_{10}$ helical motif in chain B where GSH is not bound, whereas this motif exists in a $3_{10}$ helix conformation (red) in chain A and C where GSH is bound. **b** B-factor putty representation of holo-MGST2 structure focusing on loop conformation of the $3_{10}$ helical motif at the active site without GSH. Dark red color with wider tube indicates regions with higher B-factor. **c** Stability of $3_{10}$ helical motif in apo-MGST2 MD simulations. RMSD distribution for three 500 ns MD replicate simulations. Unfolding of the $3_{10}$ helix happens only at one of the subunits at a time for the apo state as shown by rMSD increase in one chain every time (top panels, compare with Supplementary Fig. 8) and involves breaking of the H-bond between Ala101 and Arg104 (bottom panels) along with specific inter-subunit contacts with the neighboring subunit (see further details in Supplementary Fig. 8).

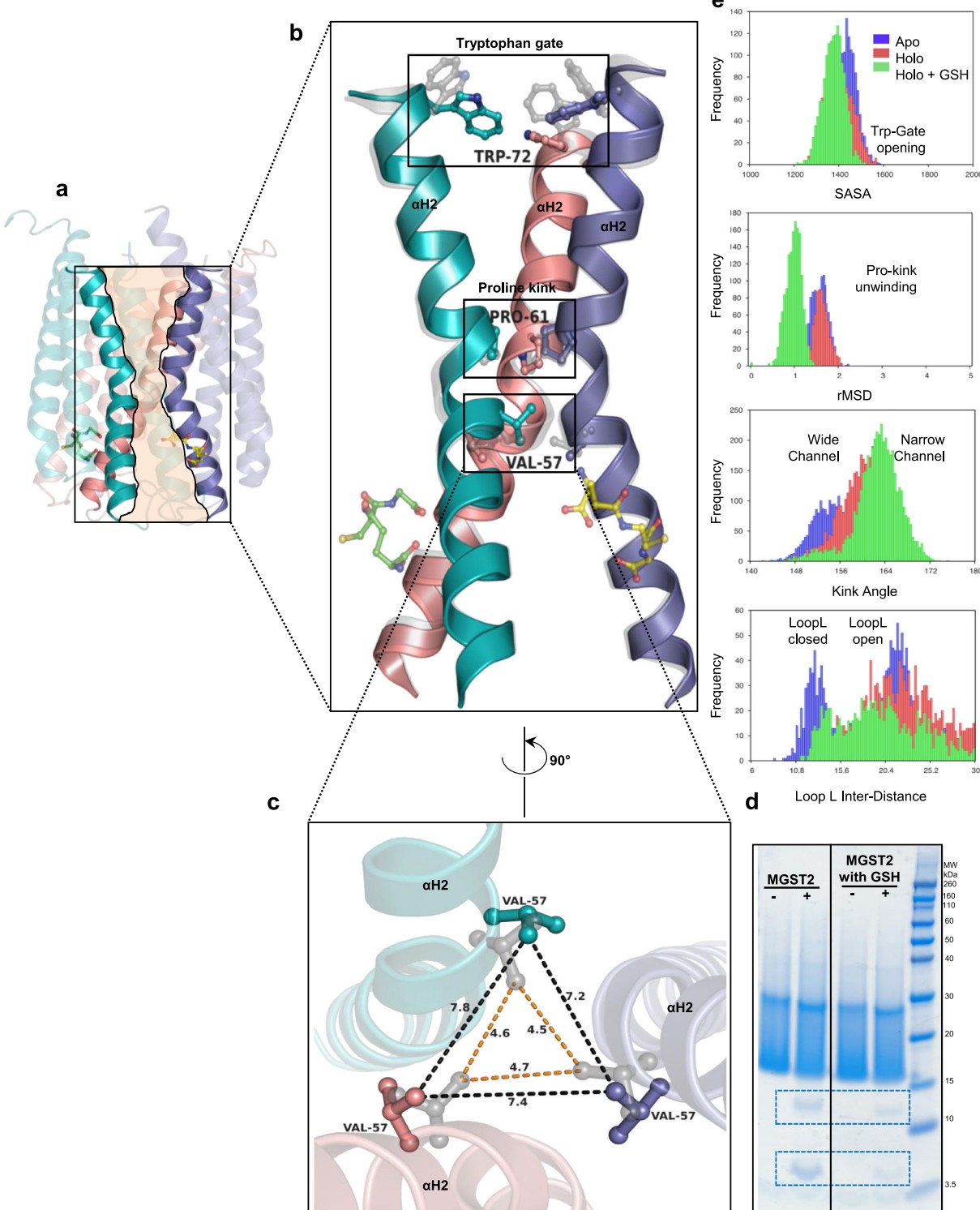

**Fig. 6 Comparison of apo- and holo-MGST2 structures unveils conformational changes of the central channel associated with co-substrate binding.**
**a** Holo-MGST2 (cartoon) with highlighted central channel (beige) created by αH2 of each monomer, which connects the cytoplasmic and luminal part of the enzyme. **b** Superimposed αH2 of holo-MGST2 (opaque cartoon) and apo-MGST2 (gray translucent cartoon) displaying dynamic residues Trp72, Pro61, and Val57 located at the trimer interface. Bound GSH molecules are depicted in ball and stick representation. **c** Widening of the cytoplasmic cone represented by a triangle between adjacent c∂-distances of Val57 in apo (gray translucent ball and stick), which increases from ~4.7 Å (orange dashed lines) to ~7.8 Å (blue dashed lines) in holo-MGST2 (opaque ball and stick). **d** Trypsin digestion of MGST2 enzyme in the presence of GSH tends to be more stable as reflected in less intensity of trypsin-digested bands compared with the enzyme digested with no substrates. Similar band pattern was observed in three independent digestions. Source data are provided as a Source data file. **e** Key dynamic descriptors along MGST2 axis in MD simulations: the Trp-gate is closed in the presence of GSH making the luminal cavity less accessible for solvent (top); pro-kink unwinding is observed in the absence of GSH as indicated by the shift in rMSD (second top); kink angle is reduced by GSH binding making the central pore narrower (third top); loop L has a marked open/close dynamics in the apo state (bottom).

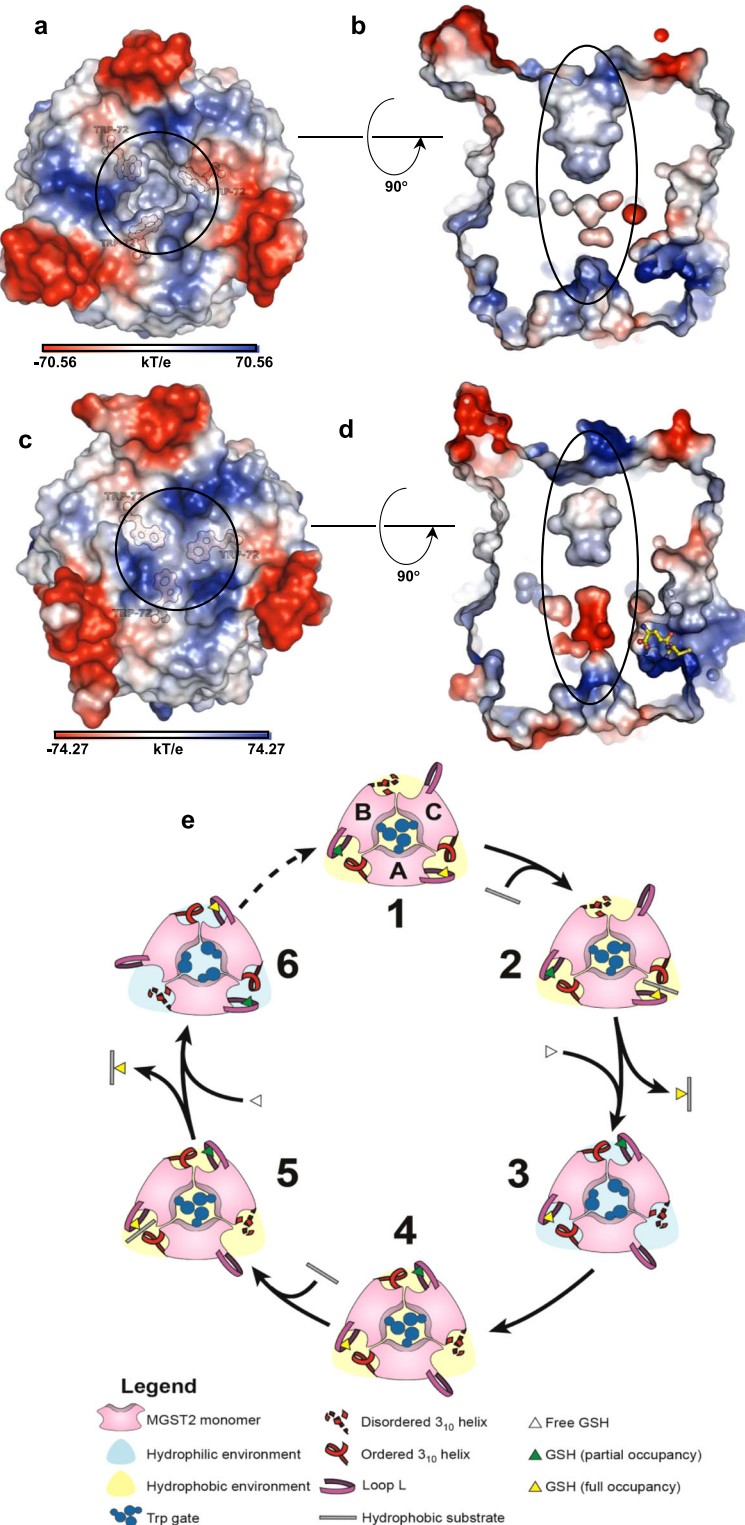

**Fig. 7 Dynamic asymmetry between the cytosolic and luminal side of MGST2 coupled to solvent accessibility and one-third of the sites reactivity.**
**a**, **c** Electrostatic surface representation of apo- and holo-MGST2 structures showing the opened and closed Trp-gate, (black circled region), respectively.
**b**, **d** Sliced through representation of **a**, **c**, respectively, along the membrane plane showing the central solvent accessible channel in black ellipsoid. **e** Proposed scheme of one-third of the sites reactivity. Cytoplasmic view of holo-MGST2 with three monomers represented as A, B, and C. (1) shows the AC site with activated GSH, AB site with partially bound GSH, and BC site empty. In AC and AB sites, the $3_{10}$ helical motif is ordered (helix conformation) and loop L at the cytosolic side is closed as a lid, while the BC site has disordered $3_{10}$ helical motif (loop conformation) and open loop L lid. The closed Trp-gate at the luminal side restricts solvent access and creates a hydrophobic environment, which promotes the entry of hydrophobic second substrate and conjugation at the AC site (2). Next the $3_{10}$ helix unfolds and loop L lid and the Trp-gate open to create a hydrophilic environment and facilitates the product release at the AC site. Simultaneously, the GSH is activated at the AB site and new GSH is partially bound at the BC site where the $3_{10}$ helical motif is now in a helical conformation and loop L lid closed (3). Finally the Trp-gate closes again (4) and the next cycle of conjugation begins at the AB site (5, 6) as described in 1–4.

**Table 1 Data collection and refinement statistics.**

| | Apo-MGST2 | MGST2-GSH | MGST2-GSO$_3^-$ | MGST2—3.8 Å |
|---|---|---|---|---|
| *Data collection* | | | | |
| Space group | P1 | C222$_1$ | C222$_1$ | C222$_1$ |
| Cell dimensions | | | | |
| $a$, $b$, $c$ (Å) | 54.88, 72.16, 72.69 | 112.28, 152.37, 71.14 | 110.42, 148.87, 70.33 | 109.94, 151.14, 70.25 |
| $\alpha$, $\beta$, $\gamma$ (°) | 67.9, 86.71, 86.86 | 90.0, 90.0, 90.0 | 90.0, 90.0, 90.0 | 90.0, 90.0, 90.0 |
| Resolution (Å) | 2.49 (2.58–2.49) | 2.49 (2.58–2.49) | 2.99 (3.10–2.99) | 3.80 (3.93–3.80) |
| $R_{merge}$ | 0.188 (1.131) | 0.116 (1.34) | 0.223 (2.139) | 0.423 (2.247) |
| $I/\sigma I$ | 5.31 (0.99) | 11.12 (1.45) | 9.36 (1.11) | 4.64 (1.15) |
| Completeness (%) | 98.29 (96.51) | 99.88 (99.34) | 99.72 (99.23) | 99.42 (99.66) |
| Redundancy | 3.5 (3.6) | 7.5 (7.7) | 9.1 (9.6) | 12.2 (12.4) |
| *Refinement* | | | | |
| Resolution (Å) | 43.33–2.49 | 46.28–2.49 | 45.26–3.0 | 44.45–3.8 |
| No. of reflections | 35,150 | 21,513 | 11,946 | 6009 |
| $R_{work}/R_{free}$ | 0.2184/0.2670 | 0.2263/0.2744 | 0.2432/0.2814 | 0.3050/0.3368 |
| No. of atoms | | | | |
| Protein | 6213 | 3221 | 3197 | 3148 |
| Ligand/ion | 121 | 82 | 69 | — |
| Water | 71 | 46 | — | — |
| *B*-factors | | | | |
| Protein | 49.30 | 59.49 | 91.70 | 142.20 |
| Ligand/ion | 53.50 | 71.58 | 98.00 | — |
| Water | 50.00 | 60.92 | — | — |
| R.m.s. deviations | | | | |
| Bond lengths (Å) | 0.005 | 0.010 | 0.016 | 0.009 |
| Bond angles (°) | 0.88 | 1.28 | 1.66 | 1.20 |
| PDB ID | 6SSS | 6SSU | 6SSW | 6SSR |

Values in parentheses are for the highest-resolution shell.

**Crystallization and X-ray data collection**. MGST2 protein was crystallized in lipidic cubic phase at concentration of 30 mg ml⁻¹. In all, 8.8 MAG (Avanti) was used to get apo-MGST2 crystals, whereas GSH and GSO$_3^-$ complex crystals were obtained with 9.9 MAG (Sigma) as host lipid (Supplementary Fig. 16). The ratio of protein/lipid was maintained as 2:3, respectively, for both lipids. DOPC (Sigma) was added as an additive lipid to the protein solution in the form of dry powder at 3 mg ml⁻¹ concentration. GSH and GSO$_3^-$ were added separately to the protein at 7 mM concentration to get the complex. The protein solution was homogenized with corresponding host lipid in a coupled syringe-mixing device at room temperature. Using Mosquito LCP robot (TTP Labtech), 50 nl of the resultant cubic phase was dispensed onto 96-well Laminex glass plate (Molecular Dimensions), covered with 800 nl of precipitant solution containing 100 mM sodium acetate pH 4.5, 17% (v/v) PEG 400, 400 mM sodium sulfate, and 100 mM potassium thiocyanate for apo crystals. The precipitant solution for GSH complex contains 100 mM MES pH 5.5, 400 mM lithium citrate, 20% (v/v) PEG 400, and 100 mM sodium nitrate and, for GSO$_3^-$, 100 mM MES pH 5.5, 400 mM lithium citrate, 20% (v/v) PEG 400, and 100 mM sodium malonate. Plates were sealed with Laminex glass cover (Molecular Dimensions) and stored at 20 °C. X-ray data were collected at i04 and i04-1 beamlines at Diamond Light Source (DLS) and ID23-2 beamline at European Synchrotron Radiation Facility (ESRF).

**Structure determination**. Diffraction data were indexed, integrated, and scaled with XDS[20]. Initial phases were obtained by molecular replacement in PHASER[21] using coordinates of human LTC4S (PDB ID: 2UUI) without heteroatoms as search model for low-resolution (3.8 Å) MGST2 structure. The 3.8 Å structure was used as a model for the other three reported structures. CHAINSAW[22] in CCP4 package[23] was used to prepare the search model for PHASER. Refinement was carried out with REFMAC[24] in CCP4 package[23] and PHENIX[25]. Model building and ligand fitting were done in COOT[26]. PyMOL was used to generate all structural figures. Statistics of final models and PDB IDs were given in Table 1.

**Determination of MGST2 topology**. Wild-type *Mgst2* gene was engineered by introducing *N*-linked glycosylation site (Asn-Ser-Thr) followed by a linker sequence (Gly-Ser-Ala-Gly-Ser-Ala-Gly-Ser-Ala-Gly) between the residues 2Ala-3Gly. The engineered *Mgst2* gene was cloned into pGEM1 vector for expression. The construct was transcribed and translated in vitro using TNT® SP6 Quick Coupled System (Promega) in the presence and absence of column-washed rough microsomes (CRM) of pancreas from dog. In all, 5 µl of reticulocyte lysate, 500 ng of plasmid DNA, 0.5 µl of [³⁵S]Met, and 0.5 µl of CRM were mixed and incubated at 37 °C for 90 min. For Endo H treatment, 9 µl of the TNT reaction was mixed with 1 µl of 10×

glycoprotein denaturing buffer, 0.5 µl of Endo H (500,000 units/ml; New England BioLabs), 7.5 µl of MillQ water, and 2 µl of 10× GlycoBuffer 3. The reaction mix was incubated at 37 °C for 1 h. Translated products were analyzed by 12% SDS-PAGE after heating at 40 °C for 10 min, and the gel was visualized on Fuji FLA-3000 PhosphorImager (Fuji film) with the Image Reader 8.1J/Image Gauge software.

**MGST2 site-directed mutagenesis**. Site-directed mutagenesis was performed to generate MGST2 mutant variants by following the QuickChange protocol (Stratagene, La Jolla, CA). The plasmid pPICZA carrying wild-type *Mgst2* gene was used as a template for PCR to generate all different mutants using respective primers (Supplementary Table 2). The positive clones were confirmed by DNA sequencing (SEQLAB, Göttingen, Germany) to verify the corresponding nucleotide changes.

**UV difference spectroscopy and analysis of pH dependence**. To observe the formation of thiolate anion at 239 nm, UV difference spectra was measured. Wild type and Arg104Ala mutant of MGST2 were pre-mixed with 0.5 mM GSH in a buffer containing 100 mM Tris-HCl pH 7.2 and 0.05% (w/v) DDM. The difference spectra were recorded on a Philips PU8720 spectrophotometer at room temperature between 200 and 400 nm by subtracting the spectra of the enzyme with GSH from the spectra of the enzyme without GSH and only buffer with GSH[7]. The thiolate anion formation was observed as an increase in absorbance at 239 nm. The addition of an inhibitor GSO$_3^-$ (20 µM) abolished the thiolate anion peak as expected.

The pH dependence of enzyme activity was measured for both wild-type MGST2 and Arg104Ala mutant at different pHs ranging from 7.0 to 9.0 using 1-chloro-2,4-dinitrobenzene (CDNB) as an electrophilic substrate. The steady-state kinetic parameters of enzyme-catalyzed conjugation reaction catalyzed by wild-type MGST2 and Arg104Ala mutant were also determined by varying CDNB (0.1–1 mM) concentration while keeping GSH concentration constant at 5 mM. The measurements were performed in triplicates, and experimental data were fit to the Michaelis–Menten equation using MMFIT and RFFIT in the SIMFIT program (http://www.simfit.man.ac.uk) to calculate kinetic parameters (Supplementary Table 1).

**LTC4 synthase assay**. The enzymatic conjugation of GSH with LTA$_4$ was monitored at 280 nm by high-performance liquid chromatography. The enzyme (0.8–1.2 µg) in 100 µl reaction buffer containing 25 mM Tris-HCl pH 7.8, 100 mM NaCl, 0.05% (w/v) DDM, and 5 mM GSH was incubated in the presence of 30 µM LTA$_4$ at room temperature, and the reaction was terminated after 15 s by the addition of 200 µl methanol. Prostaglandin B$_2$ (100/300 pmol) was then added as an internal standard to the final reaction volume.

**Circular dichroic (CD) analysis**. Far UV CD spectra of wild-type MGST2 and mutants were measured with J-810 spectropolarimeter (Jasco) with a path length of 0.01–0.05 cm (Supplementary Fig. 14).

**Trypsin digestion**. Purified MGST2 was subjected to trypsin in a 30-μl reaction mixture containing 10 μg of wild-type MGST2 protein, 30 ng of trypsin (type III from bovine pancreas, Sigma T-8253), 0.05% (w/v) DDM, 8.3 mM potassium acetate, 200 mM KCl, 6.5% (v/v) glycerol, 0.7 mM Na/EDTA, 20 mM Tris-HCl pH 8.0, and 1 mM CaCl$_2$ and incubated for 30 min at 37 °C. The reaction was terminated by the addition of 100 ng of trypsin inhibitor (type II-S from soybean, Sigma T-9128). Samples of 5 μg of protein were run on SDS-PAGE.

**Molecular dynamics (MD) simulations**. We performed unbiased MD simulations[27,28] collecting in total 1.5 μs (3 × 500 ns) for three conditions: the apo structure, the holo structure with GSH, and the holo structure after GSH removal. All-atom GSH and POPC topologies for the Gromos 54A7 force field were obtained from the automated topology builder[29,30] and repository. Proteins were placed in a POPC bilayer and the system was titrated, neutralized, hydrated, minimized, heated, and equilibrated following the state-of-the-art protocols[31] with standard conditions close to physiological ones (pH 7.4, with Na$^+$ and Cl$^-$ ions added to neutralize the system). Multiple production replica trajectories were carried out using the GROMACS[32] simulation engine (version 5.1.4) with no restraints under the NPT ensemble. All trajectory analyses were performed with in-house code in bash, python, C++, and FORTRAN as well as with Visual Molecular Dynamics[33] and GROMACS utilities to compute RMSD, distances, angles, solvent accessibility, etc. For comparative principal component analysis, shown in Supplementary Fig 10, MGST2 structures were aligned with the LTC4S trimer 2PNO to extract structural motions encoded by MAPEG family and build a conformational landscape to track simulations[34].

**Reporting summary**. Further information on research design is available in the Nature Research Reporting Summary linked to this article.

## Data availability

Atomic coordinates and structure factors have been deposited in the Protein Data Bank with accession codes 6SSR (MGST2 at 3.8 Å structure), 6SSS (apo-MGST2 structure), 6SSU (MGST2-GSH complex), and 6SSW (MGST2-GSO$_3^-$ complex). Structures used for molecular replacement and principal component analysis can be found under PDB accession code 2UUI and 2PNO, respectively. MD trajectories, analysis scripts, and other data are available from the corresponding authors upon reasonable request. Source data are provided with this paper.

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

## Acknowledgements

We are grateful to Ralf Morgenstern, David Drew, IngMarie Nilsson, and Michaela Mårback for sharing resources and advice. We thank Joseph Brock for modeling the terminal ends of MGST2 and help in preparing GSH topology and set-up for MD simulations. We also acknowledge the computational resources from the Swedish National Infrastructure for Computing (SNIC 2018/2-37) and synchrotron resource from ESRF and DLS with excellent support from beamline scientists.This work was supported by the Swedish Research Council (2018-02818, 2014-06119), the Linneus Grant CERIC, and Novo Nordisk Foundation (NNF15CC0018346 and 0064142).

## Author contributions

M.T. purified, crystallized, and solved structures of MGST2. MD simulations were carried out and analyzed by L.O. E.N. crystallized apoenzyme and provided LCP technology. S.A. did mutational and steady-state kinetics analysis. A.R.M. helped supervise the project, gave intellectual input, and revised the manuscript. M.T., L.O., and J.Z.H. wrote the manuscript and J.Z.H. conceptualized, planned, and supervised the project.

## Funding

## Competing interests

The authors declare no competing interests.
