## [Peer Review File · Nature Communications]

REVIEWER COMMENTS

Reviewer #1 (Remarks to the Author):

The authors report multiple crystal structures of Microsomal Glutathione Transferase 2 (MGST2), an important membrane-embedded enzyme that conjugates Leukotriene A4 with Glutathione. This is the first crystal structure of this particular member of the MAPEG family, and the work details multiple states of the enzyme and describes the structural features that allow the enzyme to achieve these distinct conformations. Molecular dynamics simulations reveal possible pathways between these multiple states. The data combine to support a robust model for how the enzyme cycles between these conformations such that only one active site of the trimer is functional at a time. The work is well done, and the contribution to the field of major significance.

A major strength of the paper are the illustrations, which are thoughtfully rendered to convey and support the investigators' model for one-site reactivity.

The paper is very well-written. I have only minor comments on the supplementary figures: The legend to Supl. Figures 8 does not adequately explain the image.

I was not able to view movies 2 and 3.

Reviewer #2 (Remarks to the Author):

In the manuscript "Crystal structures of human MGST2 reveal synchronized conformational changes regulating catalysis" Thulasingam et al use a combination of crystal structures, biochemical experiments and simulations to unravel the complex conformational changes that regulate catalysis in Microsomal glutathione S-transferase 2 (MGST2). Previous studies of MGST2 and other members of the MAPEG family (Membrane Associated Proteins in Eicosanoid and Glutathione metabolism) revealed that only one out of three active sites is used at a time. However, the molecular basis for a catalytic mechanism involving reactivity in one-third the sites was not clear. The crystal structures of MGST2's holoenzyme and in complex with GSH and GSO3- together with mutagenesis and molecular simulations support a highly dynamic equilibrium for the GSH binding site, with the apo 310 helix unfolding and refolding in one unit at a time and are suggestive of a conformational selection mechanism. The force field used in the molecular dynamics simulations (Gromos 54A7) has been developed almost a decade ago and does not benefit from the recent progress in force fields from the Amber, CHARMM and RSFF communities that enabled them to reproduce equally well folded and unfolded states of proteins. Thus, the balance between folded and unfolded states in the simulations might be incorrect. However, from a qualitative point of view, the observed mechanism should still be valid and it is supported by the experiments. Thus, the evidence for the interesting and revealing global mechanism proposed is quite solid. The same cannot be said about the proposed conformational selection mechanism, which is not validated by kinetics measurements nor by direct ligand binding simulations and thus a mixed conformational selection / induced fit mechanism cannot be ruled out. But a few words of caution in a revised manuscript can easily address this point. Overall, this paper sheds light on very complex and interesting mechanism, paving the avenue to the rational design of small molecules inhibiting MGST2.

Reviewer #3 (Remarks to the Author):

This is a relevant contribution on the important human enzyme MGST2. Moreover, the structure of

MGST2 has general significance because of its quaternary structure. The enzyme is a homotrimer and presents heterologous interfaces as defined by Monod, Wyman and Changeux in 1965. In enzymes, heterologous polymerization is much less common than isologous.

Main criticism: the authors present in great detail the structure of the active site in the different forms of the enzyme, but their description of the intersubunit interfaces and of the symmetry (or asymmetry) of the macromolecule in its holo and apo form is less clear. In view of the general relevance of this subject, which may be applicable to other, unrelated, heterologous oligomeric enzymes, a better description of this point is highly desirable.

Prof. Andrea Bellelli

Manuscript ID: NCOMMS-20-38465

Title: Crystal structures of human MGST2 reveal synchronized conformational changes regulating catalysis.

We thank the referees for their positive remarks on our work and constructive suggestions. Please find below our detailed point-by-point responses to each of the comments in verbatim. The reviewer's comments are in **black** and our responses are in **blue** font

Reviewer#1:

The authors report multiple crystal structures of Microsomal Glutathione Transferase 2 (MGST2), an important membrane-embedded enzyme that conjugates Leukotriene A4 with Glutathione. This is the first crystal structure of this particular member of the MAPEG family, and the work details multiples states of the enzyme and describes the structural features that allow the enzyme to achieve these distinct conformations. Molecular dynamics simulations reveal possible pathways between these multiple states. The data combine to support a robust model for how the enzyme cycles between these conformations such that only one active site of the trimer is functional at a time. The work is well done, and the contribution to the field of major significance. A major strength of the paper are the illustrations, which are thoughtfully rendered to convey and support the investigators' model for one-site reactivity. The paper is very well-written. I have only minor comments on the supplementary figures:

The legend to Supl. Figures 8 does not adequately explain the image.
I was not able to view movies 2 and 3.

We thank this reviewer for his positive critique of our work.

In response to comments by Reviewer #3 (see further below), we have updated the Supplementary Figure 8 with additional data in panel A. In this process, the legend has also been revised to adequately explain the “ Stability of the 3₁₀ helix in the holo state” now appearing in panel B.

The new updated legend of Supplementary Figure 8 now reads as follows:

“Asymmetry of intersubunit interactions and stability of 3₁₀ helical motif in MD simulations. A) Asymmetry of intersubunit interactions in apo simulations. Glu58 of the unfolded subunit forms a transient salt bridge with Arg30 of the neighboring one. On the top, distance distribution for the Glu58-Arg30 bond, with the glutamate at the unfolded subunit highlighted in red. A snapshot for replicate 3 is shown on the bottom. d). B) 3₁₀ helical unfolding in simulations starting from the holo MGST2 structure. RMSD distribution for three 500 ns MD replicate simulations of holo MGST2 in the absence (top panel) and in the presence (bottom panel) of GSH. Note how, in comparison with the apo state (Fig.5C, top panels), the 3₁₀ helical motif is more stable in the holo conformation, as shown by the absence of tall peaks at higher RMSD. Binding of GSH further stabilizes the 3₁₀ helical motif, resulting in narrow RMSD peaks close to the starting holo conformation.”

Movies 2 and 3 have been updated to MPEG format and should now be easier to visualize.

Reviewer #2:

In the manuscript "Crystal structures of human MGST2 reveal synchronized conformational changes regulating catalysis" Thulasingham et al use a combination of crystal structures, biochemical experiments and simulations to unravel the complex conformational changes that regulate catalysis in Microsomal glutathione S-transferase 2 (MGST2). Previous studies of MGST2 and other members of the MAPEG family (Membrane Associated Proteins in Eicosanoid and Glutathione metabolism) revealed that only one out of three active sites is used at a time. However, the molecular basis for a catalytic mechanism involving reactivity in one-third the sites was not clear. The crystal structures of MGST2's holoenzyme and in complex with GSH and GSO3- together with mutagenesis and molecular simulations support a highly dynamic equilibrium for the GSH binding site, with the apo 310 helix unfolding and refolding in one unit at a time and are suggestive of a conformational selection mechanism. The force field used in the molecular dynamics simulations (Gromos 54A7) has been developed almost a decade ago and does not benefit from the recent progress in force fields from the Amber, CHARMM and RSFF communities that enabled them to reproduce equally well folded and unfolded states of proteins. Thus, the balance between folded and unfolded states in the simulations might be incorrect. However, from a qualitative point of view, the observed mechanism should still be valid and it is supported by the experiments. Thus, the evidence for the interesting and revealing global mechanism proposed is quite solid. The same cannot be said about the proposed conformational selection mechanism, which is not validated by kinetics measurements nor by direct ligand binding simulations and thus a mixed conformational selection / induced fit mechanism cannot be ruled out. But a few words of caution in a revised manuscript can easily address this point. Overall, this paper sheds light on very complex and interesting mechanism, paving the avenue to the rational design of small molecules inhibiting MGST2.

We thank the reviewer for the positive remarks, constructive criticism and suggestions.

We fully agree with the referee that the classical force-field used (GROMOS54A7) was not specifically parameterized for unfolding/refolding processes, however, at the moment these simulations were done (2018), it was routinely used for simulations of lipids and especially, for non-standard molecules like GSH and others we planned to test. Being our goal to simulate MGST2 with/without its ligand in a simple lipid membrane (POPC), and not expecting unfolding events, our force-field choice was guided by this need as well as by the GROMACS-oriented clusters available for us and well-tested *in-house* protocols for Berger/GROMOS lipids. Specifically, we were inspired by the most extensive MD study of GSH/GSSG at the time, done with GROMOS54A7 and which reported good agreement with experiments (Vila-Viçosa et al, 2013: <https://doi.org/10.1021/jp401066v>). Therefore, we used GSH and POPC GROMOS54A7 topology files for GROMACS simulations from the ATB database (Stroet et al, 2018: <https://doi.org/10.1021/acs.jctc.8b00768>). Notably, recent comparative force-field studies of folding-unfolding proteins (see e.g. Kamenik et al, 2020: <https://doi.org/10.1063/5.0022135>) have found that GROMOS54A7 performs surprisingly well compared with experimental data as well as more recent force-fields. Thus, as the referee mentions, we think the essential mechanism is well supported by both crystal structures and simulations, but fully agree that, to accurately estimate the balance of folded versus unfolded states would require a comparative study including other force-fields, which is beyond the goals of the current work.

Similarly, regarding conformational selection, we apologize for the misleading statement, since our data rather points to a mixed mechanism, initiated by conformational selection but that

would require GSH binding to reach the productive conformation for the catalytic site. Our evidence for conformational selection playing a role in the initial binding of the ligand comes mostly from our MD simulations and its agreement with the crystal structure. Specifically, MD shows that the apo state spontaneously samples half of the transition pathway to the holo conformation, and vice versa (Supplementary Figure 10), reaching rMSD as low as 4.5 Angstroms (lower if the unfolded sites and flexible C-termini are excluded). Moreover, as MGST2 approaches the global conformation of the holo state, the binding site unfolds, hydrates and is further “preconfigured” to bind the ligand by approaching key residues (see response to R#3). Nevertheless, to reach the catalytic conformation definitely would require a GSH binding in the productive conformation.

We have made the following changes and additions in the main text:

Original submission, page 5, line 135-138, from “Overall, simulations support globally a conformational selection mechanism, in which the apo and holo unbound conformers spontaneously interconvert, and locally, a highly dynamic equilibrium for the GSH binding site, with the apo 3_{10} helix unfolding and refolding at one unit at a time (Supplementary Fig. 10)” to revised version, page 5, line 147-152: “*Globally, the apo simulations spontaneously approach the holo state and vice versa through large-scale changes like α H4 tilting (Supplementary Fig. 10), which are accompanied locally by the highly dynamic equilibrium of the 3_{10} helix unfolding and refolding at one unit at a time and other changes roughly preconfiguring a binding site. These evidences suggest an important role for conformational selection in the initial steps of MGST2 catalytic mechanism, which warrants further investigation*”

Revised version, page 11, line 326, we have added: “*For comparative principal component analysis, shown in supplementary figure 10, MGST2 structures were aligned with the LTC4S trimer (2PNO) to extract structural motions encoded by MAPEG family and build a conformational landscape to track simulations³³*”.

Original submission, reference 29 has been changed from “Koziara, K. B., Stroet, M., Malde, A. K. & Mark, A. E. Testing and validation of the Automated Topology Builder (ATB) version 2.0: Prediction of hydration free enthalpies. *J. Comput. Aided. Mol. Des.* 28, 221–233 (2014)” to “*Stroet, M., Caron B., Visscher K.M., Geerke D.P., Malde A.K. & Mark, A. E. Automated Topology Builder (ATB) version 3.0: Prediction of solvation free enthalpies in water and hexane. J. Chem. Theory Comput.* 14, 5834-5845 (2018)” in the revised version.

Original submission supplemental information reference “*Orellana, L., Yoluk, O., Carrillo, O., Orozco, M. & Lindahl, E. Prediction and validation of protein intermediate states from structurally rich ensembles and coarse-grained simulations. Nat. Commun.* 7, 12575 (2016)” has been moved to the main text as reference 33 in the revised version.

Original submission, page 18, line 670-672, we have changed from “(C) Unfolding of the 3_{10} helix happens only at one of the subunits at a time for the apo state (note increase in rMSD in one of the chains) (top panels) and involves interruption of the H-bond between Ala101 and Arg104 (bottom panels)” to revised version, page 18, line 673-679: “*(C) Stability of 3_{10} helical motif in apo-MGST2 in MD simulations. RMSD distribution for three 500 ns MD replicate simulations of apo MGST2. Unfolding of the 3_{10} helix happens only at one of the subunits at a time for the apo state as shown by rMSD increase in one chain every time (top panels, compare with Supplementary Fig.8) and involves breaking of the H-bond between Ala101 and Arg104*”

(bottom panels) along with specific intersubunit contacts with the neighboring subunit (see further details in Supplementary Figure 8)".

We have made the following changes to the Supplementary information.

Original submission, page 13, from "Principle components analysis : To monitor the global conformational changes, we used the principal components (PCs) analysis as described in¹. (A): PCs computed from the ensemble formed by MGST2 X-ray structures and the structurally similar LTC4S (PDB ID: 2PNO)" to revised version, page 13: "*Spontaneous sampling of apo and holo states in MD: To monitor global conformational changes in simulations, we used principal components (PCs) analysis. (A): PCs computed from the X-ray ensemble formed by MGST2 structures and the structurally similar LTC4S (PDB ID: 2PNO). PC1&2, which capture >80% of the X-ray apo-holo change, cluster separately the two conformations. Note how MD simulations from the apo state sample half of the pathway towards the holo region and vice versa*".

Original submission, page 16, from "Movement of MGST2 on membrane" to revised version, page 16: "*Transversal movement of MGST2 in MD simulations: Representative snapshots from apo (left) and holo (right) simulations*".

Reviewer#3:

This is a relevant contribution on the important human enzyme MGST2. Moreover, the structure of MGST2 has general significance because of its quaternary structure. The enzyme is a homotrimer and presents heterologous interfaces as defined by Monod, Wyman and Changeux in 1965. In enzymes, heterologous polymerization is much less common than isologous.

Main criticism: the authors present in great detail the structure of the active site in the different forms of the enzyme, but their description of the intersubunit interfaces and of the symmetry (or asymmetry) of the macromolecule in its holo and apo form is less clear. In view of the general relevance of this subject, which may be applicable to other, unrelated, heterologous oligomeric enzymes, a better description of this point is highly desirable.

We thank the reviewer for his or her thoughtful comments and suggestion for improvement.

The inter-subunit interfaces differ between the apo and holo states of MGST2 as an effect of its dynamic transitions during catalysis. The basic, symmetric, inter-monomeric interactions that help maintain the trimeric structure are now indicated in the main text and an additional panel C of Supplementary Figure 3. We have also updated the Figure 4D to better indicate the symmetric apo and asymmetric holo inter-subunit interaction.

Additionally, we reassessed the inter-subunit interactions in MD. Our simulations already showed clearly that asymmetry of inter-subunit interactions is an important feature that allows the apo state to transition to holo-like configurations and locally unfold the 3₁₀ helix in one subunit at the time. Re-analysis of our data has highlighted a transient salt bridge specifically formed between the "unfolded" subunit Glu58 and Arg30 from the neighboring one as a asymmetry that brings together binding site residues, somehow helping to "preconfigure" it. We also have updated Supplementary Figure 8 with an additional panel A showing the transient asymmetric inter-subunit interaction. See also the response to Reviewer #1.

We have made the following changes and additions to the main text.

Revised version, page 3, line 68-71, we have added “*The protomers are held together by extensive hydrogen, hydrophobic and polar interactions. Residues from α H1, α H2, α H4 and loop L, but not from α H3 are involved in the inter-subunit interactions. Specifically, Glu58 on α H2 provides a major symmetric polar interaction with Gln53 on α H2 of opposite monomer (Supplementary Fig. 3A-C)*”.

Original submission, page 5, line 123-126, we have changed from “ Strikingly, in apo simulations the 3_{10} helix at the active site unfolds in one unit at a time (Fig. 5C and Supplementary mov. 1). This local unfolding is not subunit-specific, occurring in a different monomer in each replica trajectory, and is also reversible with some runs undergoing transient refolding” to revised version, page 5, line 130-138: “*Strikingly, in apo simulations on a membrane, the symmetry of inter-subunit interactions early breaks and the 3_{10} helix at the active site unfolds in one unit at a time (Fig. 5C and Supplementary Movie. 1). Moreover, such unfolding is also reversible with some runs undergoing transient refolding. This local unfolding is not subunit-specific, occurring in a different monomer in each replica trajectory, and is accompanied by changes in inter-subunit contacts. While Glu58 interacts by forming H-bonds with both Gln53 and Gln54 in all interfaces, only in the unfolded subunit forms a transient salt-bridge with Arg30 of the neighbouring one, bringing closer GSH binding site residues (Supplementary Fig. 8A)*”

Original submission, page 17, line 613-616 we have changed from “(D) Asymmetric inter-monomeric interaction between α H2 (cartoon): Glu58-Gln54 (ball and stick) interaction (dashed lines) remains same at the GSH (stick) unoccupied and fully occupied sites whereas Glu58 switches its interaction from Gln54 to Gln53 at the partially occupied site” to revised version, page 17, line 619-623: “*(D) Asymmetric inter-monomeric interaction between α H2 (cartoon): Glu58 switches its interaction from Glu53 to Gln54 at the GSH unoccupied and fully occupied sites whereas Glu58 remains interacting with Gln53 at the partially occupied site as observed in the apo structure. Apo residues and interactions are shown in gray translucent ball and stick and orange dashed lines respectively*”.

We have made the following changes to the Supplementary information.

Original submission, page 6, we have changed from “(A) Asymmetric unit of apo MGST2 contains two molecules of trimer (Ca RMSD= 0.46.) in cylindrical representation with lipid molecules (MAG 8.8 in purple ball and stick) at the interface; (B) Cartoon rendering of MGST2 trimer with bound sulfate molecules (ball and stick) at the active site, view from the membrane plane; (C) view from the cytoplasm. The 3_{10} helical motif is highlighted in red” to revised version, page 6: *(A) Cartoon rendering of MGST2 trimer with bound sulfate molecules (ball and stick) at the active site, view from the membrane plane; (B) view from the cytoplasm. The 3_{10} helical motif is highlighted in red; (C) symmetric polar inter-subunit interactions (dashed lines) established between Glu58 and Gln53 (ball and stick) located on the α H2 of each monomer; (D) asymmetric unit of apo MGST2 contains two molecules of trimer (C α RMSD= 0.46Å) in cylindrical representation with lipid molecules (MAG 8.8 in purple ball and stick) at the interface.*

By these changes and additions, we hope we have been able to accommodate the reviewers reservations.

REVIEWERS' COMMENTS

Reviewer #1 (Remarks to the Author):

The revised manuscript thoroughly addresses my previous concerns.

Reviewer #2 (Remarks to the Author):

The authors satisfactorily addressed all the questions I had in the revised manuscript.

Reviewer #3 (Remarks to the Author):

I think that the comments of the reviewers were satisfactorily addressed and that the manuscript is worth publishing.

Manuscript ID: NCOMMS-20-38465A

Title: Crystal structures of human MGST2 reveal synchronized conformational changes regulating catalysis.

We thank the referees for their positive remarks on our work.

Reviewer#1:

The revised manuscript thoroughly addresses my previous concerns.

We thank the reviewer for the positive comment.

Reviewer #2:

The authors satisfactorily addressed all the questions I had in the revised manuscript.

We thank the reviewer for the positive comment.

Reviewer#3:

I think that the comments of the reviewers were satisfactorily addressed and that the manuscript is worth publishing.

We thank the reviewer for the positive comment.